# Maternal Periodontal Status as a Factor Influencing Obstetrical Outcomes

**DOI:** 10.3390/medicina59030621

**Published:** 2023-03-20

**Authors:** Petra Völgyesi, Márta Radnai, Gábor Németh, Krisztina Boda, Elena Bernad, Tibor Novák

**Affiliations:** 1Department of Obstetrics and Gynecology, Faculty of Medicine, University of Szeged, 6725 109 Szeged, Hungary; 2Department of Prosthodontics, Faculty of Medicine, University of Szeged, 6725 109 Szeged, Hungary; 3Department of Medical Physics and Informatics, Faculty of Medicine, University of Szeged, 6725 109 Szeged, Hungary; 4Department of Obstetrics and Gynecology, Faculty of Medicine, “Victor Babes” University of Medicine and Pharmacy, 300041 Timisoara, Romania; 5Clinic of Obstetrics and Gynecology, “Pius Brinzeu” County Clinical Emergency Hospital, 300723 Timisoara, Romania; 6Center for Laparoscopy, Laparoscopic Surgery and In Vitro Fertilization, “Victor Babes” University of Medicine and Pharmacy, 300041 Timisoara, Romania

**Keywords:** preterm birth, gingivitis, periodontitis, neonatal birth weight

## Abstract

*Background and Objectives:* Preterm birth as a complex phenomenon is influenced by numerous endogenic and exogenic factors, although its exact cause often remains obscure. According to epidemiological studies, maternal periodontal diseases, in addition to affecting general health, can also cause adverse pregnancy outcomes. Nonetheless, the existing results in the literature regarding this topic remain controversial. Consequently, our study aimed to determine the connection between poor maternal periodontal status and neonatal birth weight. *Materials and Methods*: A total of 111 primigravida–primiparous pregnant, healthy women underwent a periodontal examination in the second trimester of their pregnancies. Probing depth (PD) and bleeding on probing (BOP) were determined, and based on these diagnostic measurements, the patients were divided into three subgroups according to their dental status: healthy (H, *n* = 17), gingivitis (G, *n* = 67), and periodontitis (P, *n* = 27). *Results:* Considering that poor maternal oral status is an influencing factor for obstetrical outcomes, the presence of PD and BOP (characterized by the sulcus bleeding index, SBI) was evaluated. In the case of P, defined as PD ≥ 4 mm in at least one site and BOP ≥ 50% of the teeth, a significant correlation between BOP and a low neonatal birth weight at delivery (*p* = 0.001) was found. An analysis of the relationship between SBI and gestational age (GA) at the time of the periodontal examination in the different dental status groups showed a significant correlation between these parameters in the G group (*p* = 0.04). *Conclusions:* Our results suggest that a worse periodontal status during pregnancy may negatively affect obstetrical outcomes, especially the prematurity rate and newborn weight. Therefore, the importance of periodontal screening to prevent these complications is undeniable.

## 1. Introduction

Prematurity is a global health issue that is becoming the leading cause of newborn morbidity and mortality worldwide [1]. According to the latest data from the World Health Organization (WHO), approximately 15 million preterm births (PBs) occur worldwide each year [2]. Over the years, many risk factors have been identified that may be associated with preterm birth and low birth weight, including age, tobacco use, alcohol use, and infections [3]. In 1996, Offenbacher et al. were the first to report a possible correlation between periodontal disease and preterm birth [4], and, since then, an increasing number of studies have addressed the relationship between the two conditions [5,6]. Periodontal diseases include gingivitis and periodontitis, mainly caused by inappropriate oral hygiene or dental plaque misalignment [7].

Gingivitis, a milder and reversible pathological condition, manifests itself as inflammation of the gums due to accumulated bacterial plaque. When chronic inflammation extends from superficial to deep tissues, severe damage occurs in the tooth-supporting apparatus, which can lead to periodontitis. Initially, the clinical signs of these periodontal ailments include gingival edema and bleeding, and in the case of insufficient oral hygiene and/or regular dental care, with the deterioration of this condition and the development of periodontal pockets, tooth loss can occur [8]. Interestingly, studies have shown that pregnancy-induced hormonal changes can further aggravate gingival inflammation or the severity of pre-existing periodontal diseases [9,10]. The elevated levels of progesterone and estrogen may increase vascular permeability, making fibrous tissues more vulnerable to bacteria and resulting in adverse gingival changes [11]. With this damaging phenomenon, oral pathogens (e.g., Gram-negative microorganisms) and their bacterial products can easily reach the uterus through the bloodstream, and, at the same time, microbial components and inflammatory mediators derived from periodontal disease can also circulate to the liver, where they can initiate an entire inflammatory cascade. Consequently, prostaglandin production increases, which may cause preterm uterine activity, premature rupture of membranes, cervical insufficiency, and preterm labor [12]. Scientific data have proven that preterm delivery is strongly associated not only with a higher level of gingival prostaglandin E2 but also with increased neonatal immunoglobulin M seropositivity for several oral bacteria.

In recent studies [13], the role and the clinical and microbiological management of the oral microbiota have been described, focusing on the personalization of periodontal clinical practices and a proactive approach.

Primarily, the most reliable diagnostic tools for the identification and classification of periodontal diseases are probing depth (PD) and bleeding on probing (BOP), characterized by the sulcus bleeding index (SBI) [14]. The present study aimed to evaluate the correlation between periodontal status, based on a clinical examination of PD and BOP, and obstetrical outcomes, including neonatal birth weight (BW) and gestational age (GA) at delivery.

## 2. Materials and Methods

### 2.1. Study Design

This prospective clinical study was conducted at the University of Szeged, Department of Obstetrics and Gynecology, where, based on the authors’ previously published results, a dental unit was installed and where the examination of the selected pregnant patients was carried out. It is important to know that, in Hungary, all pregnancies that finish with deliveries are recorded in the national health system database. Dental examinations are compulsory during pregnancy, and this fact is noted in the Health Booklet, which follows an expectant woman from the start to the end of her pregnancy. A total of 111 healthy, without significant illnesses, primigravida–primiparous pregnant women were involved in this study, which was ethically approved by the Human Investigation Review Board of the Albert Szent-Györgyi Clinical Center, Szeged, Hungary, approval number (123/2019-SZTE). Patients were selected during regular pregnancy-related ultrasound examinations, performed around the gestational ages of 11 and 19 weeks. Multiple gestations or cases with any associated diseases or regular medications were excluded. The study period for the periodontal examinations was between 1 August 2019 and 29 February 2020. Szeged is the county seat of Csongrád-Csanád County, with a population of about 159,000 inhabitants. The Department of Obstetrics and Gynecology, a part of Albert Szent-Györgyi Medical School, is a state-financed unit and the only place where childbirth (about 2500/year) takes place in this city. Only inhabitants of Szeged were selected to take part in this study in order to eliminate problems related to the scheduling of periodontal examinations for pregnant patients from other places. The ultrasound examinations were performed by authorized doctors, based on a previous schedule, with the medical unit conducting about 20 similar examinations/day. This number also includes pregnant women from other cities and villages because the department is the regional tertiary medical unit for the southeastern part of Hungary, serving four counties (Csongrád-Csanád, Békés, Bács-Kiskun, and the southern part of Jász-Nagykun Szolnok), with a total population of about 1.6 million inhabitants. In the study period, there were a total of 2860 ultrasound examinations, of which 1144 were carried out on inhabitants of Szeged. Finally, of this number, 111 primigravida–primipara, healthy women were included in the project and selected for periodontal examinations. The health status of all 111 patients was monitored throughout their pregnancies; all women gave birth at our institution. The patients were fully informed about the aim of this study, and they took part voluntarily after signing a written consent form. A dental unit was installed at the Department of Obstetrics and Gynecology, based on the author’s previous studies concerning the association between maternal dental modification and obstetric outcomes, and it was used for the dental examinations. It was equipped with good lighting and the possibility of patient positioning. Gestational age was determined by carrying out sonographic measurements of the embryos in the first trimester.

Periodontal examinations were performed after the ultrasound screening, based on a previous appointment, by an experienced dentist, according to the WHO guidelines. Patients’ periodontal status was determined using PD and BOP. A disposable periodontal probe, which had a 0.5 mm diameter tip, was used for these measurements. PD was measured at 6 sites per tooth (the mesiobuccal, midbuccal, distobuccal, mesiolingual, midlingual, and distolingual sites), with the exception of the third molars, while BOP was recorded after 15 s on a Yes/No scale at the same sites as where PD was previously determined. The criteria used in each group were as follows: periodontally healthy patients had a sulcus depth between 1 and 3 mm with non-detectable gingival bleeding; patients with gingivitis had the same PD as healthy individuals, but they also had a BOP ≥ 25%; and patients with periodontitis had a PD ≥ 4 mm in at least one site and a BOP ≥ 50% of the teeth. The determination of periodontitis was based on PD and BOP, the two most important periodontal parameters. The selection of these factors was based on our previous clinical studies in this field. PD ≥ 4 mm is regarded as a “critical probing depth”, while smaller PD values are considered normal. BOP is a well-accepted sign of periodontal inflammation, and BOP and PD together are significant factors in the staging of periodontal disease [5,6,14]. After the deliveries, GA and neonatal birth weight (BW) were analyzed, and correlations regarding the patient’s previous periodontal status were explored. During the examination carried out by the dentist, in all cases, the expectant women were instructed on correct oral hygiene procedures, including toothbrush and dental floss usage. The patients were classified into 3 categories based on the results of the aforementioned measurements: 17 healthy individuals (H), 67 patients with gingivitis (G), and 27 patients with periodontitis (P). The BOP was recorded and determined for each tooth. The percentage of BOP cases was calculated according to the total number of teeth of the patient, and this approach was named the sulcus bleeding index (SBI). For example, if the total number of teeth was 28, and BOP was found in 25 teeth, the SBI was 89.28%.

Delivery was considered at-term if it occurred after the completion of the 37th week. Before this gestational age (24^+0^–36^+6^ weeks), it was noted as a preterm delivery. A low BW was defined if the newborn weight was under the 5th percentile.

### 2.2. Statistical Analysis and Graph Editing

The recruited patients were divided into three groups according to their dental status. The samples were characterised as the mean and standard deviation (SD) of the data. The group means were compared by a one-way ANOVA, which can be considered a generalisation of the Student’s two-sample *t*-test for more than two groups. The linear relationship between the examined variables was also examined by the calculation of Pearson’s correlation coefficient and the regression line. The significance of the correlation (its *p*-value) was also given. Statistical analyses were performed using SPSS version 26, and *p* < 0.05 was considered to be statistically significant. Graphs were edited in Microsoft Excel.

## 3. Results

The recruited patients were divided into three groups, namely, H, G, and P, according to the different dental statuses diagnosed during pregnancy. The low rate of cases with a healthy dental status at the time of the periodontal examination was remarkable, and this can be an important message related to the role of pre-conceptional periodontal care for women who want to have a pregnancy in the near future. All patients enrolled in this study gave birth to their newborns at the Department of Obstetrics and Gynecology, the University of Szeged.

### 3.1. The Main Gestational Age at Dental Examination, Gestational and Maternal Age at Delivery and Birth Weight at Delivery

The main gestational age at the dental examination, the gestational and maternal ages at delivery, and the birth weight at delivery are presented in Table 1. As shown, the observed difference in the birth weights of the three examined groups was found to be statistically non-significant.

### 3.2. Correlation between SBI and BW in Group P

After a further investigation of these data, it was clearly visible that the correlation was the most pronounced and statistically significant in the P group, in which a significantly lower newborn weight was observed with an increase in SBI. This means that a more severe periodontal disease was associated with a lower BW (r = −0.587, *p* = 0.001). The data are presented in Figure 1.

### 3.3. Correlation between SBI and GA at Dental Examination in Different Dental Status Groups

A detailed analysis showed a significant positive correlation (r = 0.252, *p* = 0.04) between SBI and patients’ GA at the dental examination in the G group. These detailed data are presented in Figure 2 and Table 2, where the rate of G at the time of the dental examination can be seen. Being a significant linear correlation, Figure 2 gives a more detailed picture of this relationship.

## 4. Discussion

Dental hygiene is very important during pregnancy. Preventive or diagnostic dental treatment is highly recommended at any time throughout pregnancy, along with a proper oral hygiene routine every day. Although we did not observe a connection between poor dental status and premature delivery (Table 1), our results demonstrate a significant correlation between P and a low BW at delivery (*p* = 0.001), meaning that pregnant women with a higher BOP (and, subsequently, a higher SBI) might have an increased risk of having lower BW newborns. These data are in line with the findings of Sanz et al., who previously revealed the role of hormonal and inflammatory processes in premature birth as a consequence of periodontal disease. Cytokine production and endotoxins released by oral microorganisms evoke the activation of the inflammatory cascade, which may contribute to the initiation of preterm parturition [15]. Preterm birth and a low BW may also carry a risk of predisposition to later neurological and motor impairments, together with malnutrition problems; thus, preventive medical interventions should be performed [16].

Moreover, in line with BW, gestational age at dental examination was also correlated with BOP. Interestingly, G was found to be higher at a significant rate at the time of the dental examinations (*p* = 0.04). In the context of these results, we verified that BOP had a positive correlation with the patient’s gestational age at the medical check-ups, drawing our attention to the importance of dental screening. Based on these results, it can be concluded that a worse periodontal status during pregnancy may negatively affect obstetrical outcomes. As a methodological limitation of this study, we note the relatively low number of cases. The most important cause of this was the beginning of the COVID-19 pandemic, which, at that moment, stopped study-related dental examinations. Moreover, this patient number limitation makes it impossible to differentiate between late prematurity (34^+0^–36^+6^ GW), prematurity (28^+0^–33^+6^ GW), and extreme prematurity (24^+0^–27^+6^ GW) and examine their correlations with maternal dental status. Still, regarding these causes, the results of our study have a powerful message, illustrating the importance of oral health during pregnancy.

Similar to our findings, Cho et al. found that pregnant women with dental caries participated in screenings less frequently [17]; however, clinical trials support the fact that preventive dental care can play a major role in the improvement of birth outcomes [18]. Both quantitative and qualitative studies regarding this issue have corroborated that women may have fears of dental examinations during pregnancy [19]. Although a high proportion of pregnant women experience dental problems (e.g., gingival bleeding, dental caries, and tooth mobility), only 30–40% seek medical advice during pregnancy. There may be many factors behind this pattern, but most of the time, it is due to a previous negative experience or a lack of education regarding oral hygiene. Since there is a strong correlation between poor dental status and pregnancy outcomes, raising awareness of oral health is indispensable [20]. Preventive dental and periodontal care, especially before and, if necessary, during pregnancy, is fundamental to maintaining both mothers’ and newborns’ health, and, for this exact reason, effective collaboration between obstetricians and dentists is essential [21,22]. Oral health promotion and education about proper dental care in pregnancy are indispensable for the prevention of pregnancy-associated complications, such as prematurity and neonatal low BW.

However, Scribante et al. [13] focused on the role of the balance of oral dysbiosis. They stated that using probiotics and paraprobiotics can lead to a statistically significant reduction in oral pathogenic bacterial load.

## 5. Conclusions

Periodontal diseases, such as gingivitis and periodontitis, mainly occur due to inappropriate oral hygiene. Pregnancy-induced hormonal changes further aggravate pre-existing periodontal diseases and can ultimately lead to adverse pregnancy outcomes. Our results indicate that there is a connection between periodontal status during pregnancy and obstetrical outcomes. We, therefore, believe that periodontal health programs linked to maternity are crucial for the prevention and diagnosis of periodontal diseases in pregnant women. By means of oral health education, a higher proportion of pregnancy and birth-related complications could be prevented.

Based on the obtained results, we can consider that poor maternal oral status is an influencing factor for worse obstetrical outcomes. In particular, the dental modifications associated with hemorrhage of the gingiva and periodontal pockets are involved. These modifications are detectable as early as the first trimester, and they can influence obstetrical outcomes, resulting, for example, in a lower newborn weight.

## Figures and Tables

**Figure 1 medicina-59-00621-f001:**
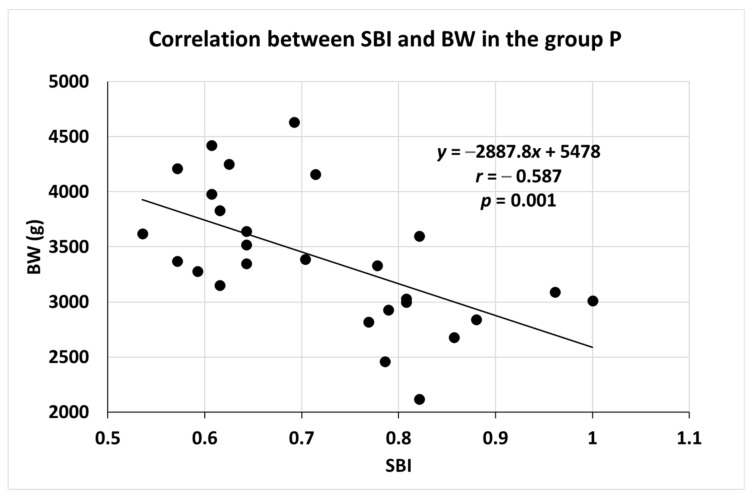
Correlation between SBI and BW in group P. Results are shown as means (±SD). BW = birth weight; SBI = sulcus bleeding index; P = periodontitis; *n* = 111; and *p* < 0.05 was considered statistically significant.

**Figure 2 medicina-59-00621-f002:**
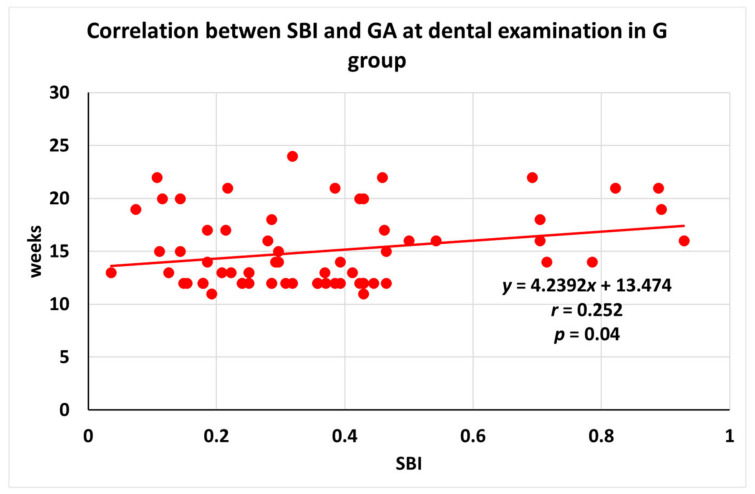
Correlation between SBI and GA at dental examination in the G group. Results are shown as means (±SD). G = gingivitis; GA = gestational age; SBI = sulcus bleeding index; *n* = 111; and *p* < 0.05 was considered statistically significant.

**Table 1 medicina-59-00621-t001:** Sample characteristics in different dental status groups. Results are shown as means (±SD).

	Healthy (H) *n* = 17	Gingivitis (G) *n* = 67	Periodontitis (P) *n* = 27	ANOVA
Gestational age at dental examination (w)	16.24 ± 3.492	15.00 ± 3.516	14.00 ± 3.150	*p* = 0.111
Gestational age at delivery (w)	38.53 ± 1.772	39.07 ± 1.454	38.67 ± 1.414	*p* = 0.253
Maternal age at delivery (y)	31.25 ± 4.337	30.91 ± 5.197	29.53 ± 6.309	*p* = 0.464
Birth weight (g)	3518.82 ± 548.212	3402.24 ± 541.443	3396.67 ± 611.826	*p* = 0.726

Analysis of invariance: *ANOVA*.

**Table 2 medicina-59-00621-t002:** Correlation between SBI and GA at periodontal examination in different dental status groups; *p* < 0.05 was considered statistically significant.

Group	N	Correlation Coefficient	*p* Value
Healthy	17	r = −0.391	*p* = 0.120
Gingivitis	67	r = 0.252	*p* = 0.04
Periodontitis	27	r = 0.127	*p* = 0.527
Total	111	r = −0.035	*p* = 0.714

## Data Availability

All data used to support the findings of this study are included within the article.

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
