# Peer review of "Maternal Periodontal Status as a Factor Influencing Obstetrical Outcomes"

_medicina, 2023, doi:10.3390/medicina59030621_

Round 1

Reviewer 1 Report

Manuscript of considerable interest for the dental sector, needs a major revision.

Abstract, to better highlight the results obtained.

Keywords, few and not present on MeSH, add more

Introduction: how does the oral microbiota of pregnant women change? what are the proactive systems to maintain a balanced microbiota? Prof Scribante's research group worked hard on this aspect.

Materials and methods: poorly described, expand them

Very confusing results, reorganize them by highlighting the results obtained

Discussion: Add proactive action using natural substances as goals.

Conclusions: rephrase them based on the comments

Bibliography: add references required

Author Response

Dear Reviewer,

Thank you very much for your very welcoming substantial recommendations and comments. Modifications and answers are included.

In the name of the authors,

Sincerely, Tibor Novak

Manuscript of considerable interest for the dental sector, needs a major revision.

Answer:

The authors considered that during the pregnancy the maternal dental associated comorbidities are directly connectedand can influence obstetrical outcome. In time recognition of these health conditions, no matter how minor apparently are, can worsen during the time of pregnancy. The article proposed for publication emphasizes the role of early detection of deterioration of the maternal dental status during pregnancy, underlying the connection between the grade of bleeding on probing and neonatal birth weight. Targeting this Special Issue is considered essential for the wide popularization of the information between the obstetrical specialists implicated in care of pregnancy.

Abstract, to better highlight the results obtained.

Answer:

Replacing the previous Results chapter with the text below (line 31)

Considering that poor maternal oral status is an influencing factor for obstetrical outcome, the presence of PD and BOP (characterized by sulcus bleeding index, SBI)were evaluated. In case of P, defined as PD4 mm at least at one site, BOP 50% of the teeth, significant correlation was found between BOP and low neonatal birth weight at delivery (p=0.001).Analysis of the date between SBI and gestational age (GA) at periodontal examination in different dental status groups,showed significant correlation between these parameters in the G group (P=0.04).

Keywords, few and not present on MeSH, add more

Answer:

We propose the next additional keywords: low birth weight, preterm delivery, maternal dental modification

Introduction: how does the oral microbiota of pregnant women change? what are the proactive systems to maintain a balanced microbiota? Prof Scribante's research group worked hard on this aspect.

 Answer: text included (line 69) with new Reference numbering

In recent studies also the role or clinical and microbiological management of the oral microbiota were described, focusing on the personalization of periodontal clinical practice and a proactive approach.

Materials and methods: poorly described, expand them

 Answer: delete previous lines 106-107, and include the text below.

The patients were fully informed about the aim of the study and they took part voluntarily, after signing the written consent form. A dental unit installed at the Department of Obstetrics and Gynecology, based on the author’s previous studies concerning the association between maternal dental modification and obstetric outcomewas enabled and used for the dental examination having with lightening and possibility for the positioning of the patients. The gestational age was determined by sonographic measurement of the embryo in the first trimester.

Very confusing results, reorganize them by highlighting the results obtained

Answer:

Modifications were made, previous 3.2, Figure 1were deleted

Discussion: Add proactive action using natural substances as goals.

 Answer: recommended reference included (line 247)

On the other hand, Scribante et al (reference nr) focuses on the role of the balance of the oral dysbiosis. They state that sing probiotics and paraprobiotics can lead to a statistically reduction oforal pathogenic bacterial load.

Conclusions: rephrase them based on the comments

Answer (additional comments included, Line 256)

Based on the obtained results we can consider that poor maternal oral status is an influencing factor for worse obstetrical outcome. Especially the dental modifications associated with haemorrhage of the gingiva or from the periodontal pockets are involved. These modifications are detectable already in the first trimesters, and can influence the obstetrical outcome including lower newborn weight.

Bibliography: add references required

Answer:

Attached

Reviewer 2 Report

I had the opportunity to review the manuscript Maternal periodontal status as a factor influencing obstetrical outcome. Although the subject is interesting and important, there are several limitations with this study that precludes its publication in the present form.

1. Introduction fails to present a gap in knowledge that could be filled by the study. The study question or hypothesis is not clearly stated. 

2. Most of the statistical analyses, especially the subgroup analysis, are no more than exploratory. Maybe spurious, since it looks that a lot of comparisons were made and the ones with p<0.05 were given attention.

3. "Almost significant" means not significant. The authors should not emphasize these results. Especially with a low sample size.

4. Discussion is superficial. It fails to present an in-depth analysis of the importance of their findings.

5. English writing needs extensive revision.

Based on these issues, I cannot recommend the study for publication.

Author Response

Dear Reviewer,

Thank you very much for your very welcoming substantial recommendations and comments. Modifications and answers are included.

In the name of the authors.

Sincerely, Tibor Novak

I had the opportunity to review the manuscript Maternal periodontal status as a factor influencing obstetrical outcome. Although the subject is interesting and important, there are several limitations with this study that precludes its publication in the present form.

  1. Introduction fails to present a gap in knowledge that could be filled by the study. The study question or hypothesis is not clearly stated.

 Answer:

Aim of the study was modified (Line 26).

  1. Most of the statistical analyses, especially the subgroup analysis, are no more than exploratory. Maybe spurious, since it looks that a lot of comparisons were made and the ones with p<0.05 were given attention.

Answer:

The implicated results were deleted.

  1. "Almost significant" means not significant. The authors should not emphasize these results. Especially with a low sample size.

Answer:

Implicated data and Figure were excluded.

  1. Discussion is superficial. It fails to present an in-depth analysis of the importance of their findings.

Answer: new considerations were included in Discussion and also in Conclusions chapters.

  1. English writing needs extensive revision.

The paper was checked by the MDPI English Editing Service.  

Based on these issues, I cannot recommend the study for publication.

Round 2

Reviewer 1 Report

The manuscript has been properly revised, it can be published

Author Response

The article was checked for English using the software on the mdpi platform. Thank you for your suggestions that were very helpful.